# OpenReview forum: "Do as We Do, Not as You Think: the Conformity of Large Language Models"
_ICLR.cc/2025/Conference — ICLR 2025 Oral_

### Official Review · Reviewer_e696 · 2024-10-20

**Soundness:** 3
**Presentation:** 4
**Contribution:** 3
**Rating:** 6
**Confidence:** 3

**Summary:**

The paper investigates the phenomenon of conformity in large language model (LLM)-driven multi-agent systems, introducing BENCHFORM, a benchmark designed to assess conformity in collaborative AI environments. Through experiments with various LLMs using specific interaction protocols and metrics like conformity rate and independence rate, the study examines the existence of conformity, factors influencing it (such as number of rounds of interaction and majority size), and explores mitigation strategies like enhanced personas and reflection mechanisms.

**Strengths:**

I want to preface by saying that conformity is an interesting subject with many potential future applications to multi-agent systems, so I really like the problem statement and scope. That being said, I think this paper's strengths lie on the following:
1. **Clear presentation and writing**: The paper is very easy to understand and follow. The plots and tables are easy to interpret and aesthetically pleasing. Overall, I enjoyed reading this paper.
2. **Benchmark design**: The evaluation protocols and metrics are carefully designed to measure the level of conformity. This makes the experiments easy to interpret and believe.
3. **Novelty and applications**: To my knowledge this is the first paper that explores the problem of conformity, which is not unique to multi-agent systems. Current LLMs seem to have a tendency to conform to the input of their user even when they might be correct in their original statements. Moreover, in a world where LLMs are being integrated into everyday-decision making, these models might have to cooperate with other systems and humans to reach solutions to specific problems. In both of these contexts, conformity is relevant and useful to ensure correct behavior.
4. **Ablation study**: The ablation study explores ineresting dimensions that might affect the level of conformity of different LLMs, depending on the interaction time and the size of the conforming majority.

**Weaknesses:**

My main concerns with this paper, which are few, lie mainly in the experimental part. More specifically:
1. **Overt dependency on BBH**: The fact that all the reasoning questions for the conformity protocols are questions extracted from the BIG-Bench Hard (BBH) dataset might be problematic for a few reasons. First, there might be inherent biases in the resulting sub-sampled dataset that make it hard to know whether the findings are general enough. Second, conformity might be dependent on the task that is being executed, so it leaves the question of what happens when the tasks are not reasoning-intensive unanswered. Third, this significantly diminishes the contribution, as it could be interpreted as derivative work (I do not personally believe this). A simple solution would be to combine questions from different reasoning datasets, and even introduce a subset of your own.
2. **Experimental consistency**: Conformity experiments are conducted for 11 different models, however the ablation study, the behavioral study and the mitigation strategies experiments are conducted in only two of the models (Llama3-70B, Qwen2-72B). I would have been more forgiving in this respect if the experiments have been conducted in the models with best performance on the conformity metrics (GPT 4o, Llama3.1-405B). In the current state the choice of models for the ablation study, behavioral study and mitigation strategies seems arbitrary and leaves many unanswered questions regarding conformity of the stronger models.
3. **Lack of baseline**: Why didn't the authors attempt to fine-tune a smaller model on BENCHFORM? Fine-tuning a model for this task will give a clear signal of the difficulty of the benchmark, therefore also the potential for future research that uses BENCHFORM .

I would be willing to increase my score if some/most of these concerns are addressed.

**Questions:**

1. Why do you think the independece rate of Gemma2 degrades with increased number of parameters (Figure 4)?

---

> ### Author Response · Authors · 2024-11-22
>
> We thank reviewer e696 for the valuable time and constructive feedback. We provide point-to-point response below.
>
> Q1: Overt dependency on BBH.
>
> A1: Thanks for the insightful comment. After an investigation, we plan to incorporate the MMLU-Pro dataset [ref1], a benchmark encompassing 14 tasks across various subjects.  Specifically, we will select 200 questions from each task category, resulting in a total addition of 2,800 questions. Due to the urgency of the rebuttal period, modifying BenchForm and rerunning all experiments are infeasible. We are committed to addressing this in future work.
>
> Q2: Clarify why Llama3-70B and Qwen2-72B are chosen in Sec. 4 & Sec. 5.
>
> A2: Qwen2-72B is selected because: i) It achieved the highest IR score among the LLMs we tested. ii) An LLM with high IR may demonstrate more varied responses that are not influenced by others' answers, which may facilitate the ablation and behavioral studies. iii) LLMs with higher IR may exhibit unexpected or emergent behaviors, which might spark novel behavior discovery and even lead to discovering new research directions. The selection of Llama3-70B as a comparative baseline is motivated by its similar parameter scale to Qwen2-72B (70B *vs* 72B) and its established position as a widely-used LLM. We have included the reasons for choosing Qwen2-72B and Llama3-70B in the main paper:
>
> > *Qwen2-72B is chosen for study due to its achieving the highest IR, which could exhibit diverse or even unexpected responses, potentially aiding in ablation and behavioral studies. Llama3-70B is also selected for comparison, given its comparable scale and widespread usage.*
> >
>
> Q3: Experiments for more LLMs.
>
> A3: For GPT-4o, we have completed the additional experiments in Appendix §G. The overall findings are consistent with that of the submission. For Llama3.1-405B, due to its slow inference speed, we have currently finished the ablation study and the behavioral study. The remaining experiments of mitigation strategies will be provided later during the rebuttal phase. In addition, we will include the results of Glm-4-Plus [ref2], an LLM comparable to GPT-4o.
>
> Q4: Lack of baseline.
>
> A4: Good suggestion! We plan to fine-tune Llama3-8B using LoRA [ref3]. Due to the urgency of the rebuttal period, we are focusing solely on generating the data for fine-tuning data using the Wrong Guidance protocol, with each task category split into training and testing sets at a 3:1 ratio. The results will be provided later in Appendix §H during this phase. Generating the data for fine-tuning using all protocols will be addressed in future work.
>
> Q5: Clarify why Gemma2's IR drops as parameters increase.
>
> A5: The decrease in Gemma2's IR as parameters increase can be explained primarily by the observation that Gemma2-27B's performance on reasoning and understanding tasks remains comparable to Gemma2-9B, despite its larger parameter count. This finding is supported by two pieces of evidence. **Firstly**, in Gemma2's paper [ref4], Gemma2-27B shows only marginal improvements over Gemma2-9B on reasoning and language understanding benchmarks, including ARC-e, PIQA, SIQA, and BoolQ (see Rows 12-15 of Table 13, Page 7 of [ref4]). **Secondly**, both LLMs achieve identical accuracy scores of 62.4% on BenchForm (refer to Table S3 in our appendix). In addition, Gemma2-27B demonstrated higher susceptibility to the Doubt protocol, as evidenced by its higher conformity rate ($CR^D$ = 66.1%) compared to Gemma2-9B's 54.1%. These findings collectively explain why Gemma2-27B exhibits a lower IR compared to Gemma2-9B.
>
> We appreciate again your thoughtful review and we hope we addressed your concerns. Please let us know if you'd like any further information.
>
> [ref1] Mmlu-pro: A more robust and challenging multi-task language understanding benchmark, NeurIPS 2024.
>
> [ref2] Chatglm: A family of large language models from glm-130b to glm-4 all tools, arXiv 2024.
>
> [ref3] LoRA: Low-Rank Adaptation of Large Language Models, ICLR 2022
>
> [ref4] Gemma 2: Improving Open Language Models at a Practical Size, arXiv 2024.

---

> > ### Comment · Reviewer_e696 · 2024-11-22
> >
> > I appreciate the effort done by the authors to improve the manuscript with some of my suggestions and those of other reviewers. I think the reply above along with the changes made to the revision address my concern about the experimental consistency of this work by answering most of my questions regarding selection criteria of the LLMs used in the the ablation study, the behavioral study and the mitigation strategies experiments. However after having more time to reflect about the contribution of this work I have come with some other concerns that I would like to get the author's perspectives on. In particular I am concerned about the hardness of this task. I believe that a good baseline should be challenging enough to motivate research in the field. Moreover this research should aim towards goals with practical utility. So having established this, I think the problem of conformity can be easily solved by ignoring the context that is not relevant to the question, i.e. the answers provided by other agents (my suspicion is that this is what a trained model would learn). So my question is two-folded: (1) how difficult do you think the problem of conformity is, and (2) what practical scenarios do you anticipate in which a solution like ignoring the other agent's responses is unfeasible?

---

> > > ### Author Response · Authors · 2024-11-23
> > >
> > > We thank the reviewer for the insightful comments and the opportunity to clarify our contribution. Below, we address your concerns about the conformity problem:
> > >
> > > Q6: The difficulty of conformity
> > >
> > > A6: We would like to point out three points:
> > >
> > > - **Our benchmark is challenging**.  As reported in Table S3, even state-of-the-art LLMs such as GPT-4o and Llama3.1-405B struggle with the tasks under the Raw protocol (i.e., GPT-4o: 72.3%; Llama3.1-405B: 74.1%). Furthermore, as highlighted in Finding I in Sec. 3.3, all evaluated LLMs exhibit significant conformity. This conformity is particularly evident in ∆T (i.e., GPT-4o: 22.6%; Llama3.1-405B: 2.5%) and ∆D (i.e., GPT-4o: 13.0%; Llama3.1-405B: 30.2%).
> > > - **Conformity is a core aspect of LLMs.** Results from the Asch conformity experiments [ref1-3] demonstrate that humans have a strong tendency to conform. Since LLMs are trained to align with human preferences and values, they naturally inherit human tendencies, including conformity [ref4-5].
> > > - **Ongoing experiment about fine-tuning.** We are currently fine-tuning Llama3-8B using data generated under the Wrong Guidance protocol and plan to report results during the rebuttal phase. In addition to testing under the Wrong Guidance protocol, we will evaluate the LLM under the Correct Guidance protocol to determine whether it has learned to disregard or exclude wrong answers from other agents.
> > >
> > > Q7: Scenarios where ignoring other agent's responses is unfeasible.
> > >
> > > A7: We would like to emphasize the following points:
> > >
> > > - **Defining "ignorable" content is complex.** It is not straightforward to determine when the content provided by other agents can or should be ignored. For instance, in domains such as scientific research or medical diagnosis, synthesizing multiple expert opinions is often crucial rather than dismissing them outright.
> > > - **Collaboration in multi-agent systems.** In multi-agent systems, agents are often designed to achieve collective goals through group discussion, reasoning, and argumentation. However, if agents ignore each other's opinions and responses, they cannot truly collaborate. This would reduce them to isolated single-agent models, defeating the purpose of a multi-agent system.
> > > - **Conformity as a double-edged sword.** As noted by reviewer EboB and kmnc, conformity is a double-edged sword. Appropriate levels of conformity can facilitate learning from group interactions, enhancing collaborative potential.
> > >
> > > We hope these clarifications address your concerns and highlight the broader significance of this work. We greatly appreciate your continued engagement and valuable feedback!
> > >
> > > [ref1] “Effects of group pressure on the modification and distortion of judgments”, Groups, Leadership and Men: Research in Human Relations 1951
> > >
> > > [ref2] “Opinions and social pressure”, Scientific American 1955
> > >
> > > [ref3] “Studies of independence and conformity. A minority of one against a unanimous majority”, Psychological Monographs 1956
> > >
> > > [ref4] “Language models are few-shot learners”, arXiv 2020
> > >
> > > [ref5] “Training Language Models to Follow Instructions with Human Feedback”, NeurIPS 2022

---

> > > > ### Comment · Reviewer_e696 · 2024-11-23
> > > >
> > > > I think this answers most of my questions; however, I am still unsure about the way in which the conformity protocols are designed in this work. To be more specific, while it is true that complex tasks might require collaboration among multiple agents and that debate or persuasion could serve as a route to generate crucial information that might influence an agent's decision to conform or not, the current protocols in this work only condition on the responses provided by the other agents (without debate or other forms of information exchange). Therefore, there is no crucial information that could be inferred from the protocols that may positively or negatively influence an agent's answer to the task, unless there is some prior knowledge about the distribution of the other agents' capabilities for the current task.
> > > >
> > > > For this reason, I was suggesting that, under the current protocols of this work, the context provided by the other agents could potentially be ignored, and the agent could just condition on the prompt of the original task. I would be curious to understand the author's position on this point, especially if there is anything I might be misrepresenting or misunderstanding.

---

> > > > > ### Author Response · Authors · 2024-11-25
> > > > >
> > > > > We appreciate your continued thoughtful engagement with our work.
> > > > >
> > > > > Q8: The rationale for the design of current protocols
> > > > >
> > > > > A8: Our protocols follow the same design as Asch conformity experiments [ref1-3], where participants were presented with others’ answers without engaging in discussion or argumentation. In other words, the task itself is to investigate whether other agents' decisions can affect the subject agent’s decision-making. While we acknowledge that debate or persuasion could serve as a route to generate crucial information and reflect genuine reasoning, our experimental results under the current setting yielded several conclusions based on statistically significant results. We also agree that exploring additional interaction protocols, such as debate or persuasion, is a right direction. So we mention this as a part of our future work (L517-520):
> > > > >
> > > > > > … future work will focus on i) … and ii) exploring more interaction protocols that better mimic real-world collaborative environments, such as discussing answers, reasoning as a group, engaging in argumentation, etc.
> > > > > >
> > > > >
> > > > > Following Asch conformity experiments, we conduct experiments in a controllable environment to ensure meaningful evaluation and benchmarking. In terms of the “prior knowledge about the distribution of the other agents' capabilities,” this work hypothesizes that it is known and can be controlled by providing correct/wrong answers. We agree that in real-world deployment, such knowledge is unknown and uncontrollable. Therefore, we add the relevant discussion in the Limitation section (L501-504) about how the current protocols may not fully capture the nuanced dynamics:
> > > > >
> > > > > > Specifically, BenchForm and the proposed interaction protocols are crafted to identify instances where agents exhibit conformist tendencies under the influence of majority opinions. This setup provides a necessary, yet not sufficient, test for conformity, as it may not fully capture the nuanced dynamics at play when agents interact over varying contexts and with diverse inputs.
> > > > > >
> > > > >
> > > > > Q9: Why can't the context provided by other agents simply be ignored?
> > > > >
> > > > > A9:  As detailed in A7, defining "ignorable" content is complex. This is evidenced by the experiment on Llama3-8B fine-tuning (Appendix §H). We list the results here for your convenience:
> > > > >
> > > > > | Model | $Acc^C$ | $Acc^W$ |
> > > > > | --- | --- | --- |
> > > > > | Llama3-8B | 85.1% | 24.1% |
> > > > > | Llama3-8B-Finetuned | 18.7% | 90.8% |
> > > > >
> > > > > As seen, after fine-tuning, Llama3-8B’s $Acc^C$ drops significantly from 85.1% to 18.7% and its $Acc^W$ increases from 24.1% to 90.8%. The results suggest that i) the fine-tuned Llama3-8B **does NOT learn to ignore other agents’ answers**, but to exclude them. ii) It is hard for LLMs to adaptively select useful contexts and filter irrelevant ones, regardless of whether fine-tuning is applied.
> > > > >
> > > > > We hope these responses address your concerns and welcome further discussion on these points.
> > > > >
> > > > > [ref1] “Effects of group pressure on the modification and distortion of judgments”, Groups, Leadership and Men: Research in Human Relations 1951
> > > > >
> > > > > [ref2] “Opinions and social pressure”, Scientific American 1955
> > > > >
> > > > > [ref3] “Studies of independence and conformity. A minority of one against a unanimous majority”, Psychological Monographs 1956

---

> > > > > > ### Comment · Reviewer_e696 · 2024-11-25
> > > > > >
> > > > > > I don't mean to be disrespectful, but I think the authors are drawing the wrong conclusion from this experiment. Since the finetuning is done using **only the wrong guidance protocol**, it effectively shows that it is easy to change the LLM's original conformity bias with fine-tuning. Now because it is trained only in the wrong guidance protocol, it is expected that the best resulting behavior is to exclude the other agent's answers, since it maximizes accuracy. In the case that this LLM was finetuned with equal amounts of data from all protocols then again there is no information gain from the context of other agent's answers, and it is likely to ignore them. Overall, I think this is evidence that we should not adapt protocols from psychology to LLMs without careful consideration first. Humans suffer of social pressure to conform, which does not map to LLMs in any reasonable form.

---

> > > > > > > ### Author Response · Authors · 2024-11-28
> > > > > > > **Response to Reviewer e696 (1/2)**
> > > > > > >
> > > > > > > We sincerely appreciate your detailed observations and critical engagement.
> > > > > > >
> > > > > > > Q10: Clarify our conclusions.
> > > > > > >
> > > > > > > A10: We respectfully disagree that our conclusions are incorrect, though we agree with aspects of your critique. As you noticed, the fine-tuning process under the Wrong Guidance protocol demonstrated LLM‘s learning ability to exclude answers, this does not conflict with our conclusions in A9:
> > > > > > >
> > > > > > > - “The fine-tuned Llama3-8B does NOT learn to ignore other agents’ answers, but to exclude them. ” In other words, the LLM learns the pattern in the context and excludes other agents’ answers. This is consistent with your observation.
> > > > > > > - “It is hard for LLMs to adaptively select useful contexts and filter irrelevant ones, regardless of whether fine-tuning is applied.” Without fine-tuning, LLMs exhibit a strong tendency towards conformity, as discussed in Finding I. Even after fine-tuning, LLMs are unable to effectively utilize information under the Correct Guidance protocol. Instead, fine-tuning primarily instills protocol-specific behavior, such as excluding other agents' answers, but does not resolve the underlying conformity bias.
> > > > > > >
> > > > > > > We suppose there may be a misunderstanding here and would welcome further discussion.
> > > > > > >
> > > > > > > Q11: Results of fine-tuning using all protocols.
> > > > > > >
> > > > > > > A11: We have conducted additional fine-tuning experiments using data from all protocols. The results are as follows:
> > > > > > >
> > > > > > > | Model | $Acc^C$ | $Acc^W$ | $Acc^T$ | $Acc^D$ |
> > > > > > > | --- | --- | --- | --- | --- |
> > > > > > > | Llama3-8B | 85.1% | 24.1% | 38.9% | 24.6% |
> > > > > > > | Llama3-8B-Finetuned | 72.9% | 85.7% | 88.9% | 87.5% |
> > > > > > >
> > > > > > > After fine-tuning, Llama3-8B performs well in $Acc^W$, $Acc^T$ and $Acc^D$ but worse in $Acc^C$ on BenchForm. However, this does not indicate that the fine-tuned LLM has learned to ignore other agents' answers; rather, it suggests that the fine-tuned LLM has likely **learned the patterns embedded in the protocols**. This is supported by the following points: **i)** the lower $Acc^C$ compared to $Acc^W$suggests that the fine-tuned Llama3-8B may still feel influenced by others' answers, and is more inclined to exclude others’ answers. **ii)** The high $Acc^T$ and $Acc^D$, on the other hand, are due to **clear cues embedded in the prompts** for the trust and doubt protocols (e.g., ------ begin of history ------ and ------ end of history ------), which make these protocols easy for LLMs to recognize.
> > > > > > >
> > > > > > > While fine-tuning can improve performance in $Acc^W$, $Acc^T$ and $Acc^D$ on BenchForm, we emphasize that it is **NOT** **a general or ultimate solution** to BenchForm, let alone the broader issue of conformity. Instead, it shifts the LLM’s behavior to match specific protocol objectives. In real-world deployment, agents encounter complex patterns rather than predictable ones. In such cases, the fine-tuning strategy would no longer be applicable.
> > > > > > >
> > > > > > > Q12: Discussion of the behavior of ignoring other agents’ answers.
> > > > > > >
> > > > > > > A12: we would like to emphasize three points:
> > > > > > >
> > > > > > > - As previously discussed, our experiments about fine-tuning show that LLMs do NOT **learn to ignore** other agents' answers but instead **learn patterns** embedded in protocols. Current attempts do not enable LLMs to ignore other agents' answers. Even if agents completely ignore each other's responses, it would only reduce the multi-agent system to isolated single-agent models, defeating the purpose of collaboration in such systems.
> > > > > > > - Conformity, as noted by reviewers EboB and kmnc, and as acknowledged by the cognition community [ref1-2], is a double-edged sword (we also give a detailed discussion about this point in L485-488). While it can facilitate learning from group interactions and enhance collaboration, excessive conformity may hinder independent decision-making and introduce bias.
> > > > > > > - An ideal LLM can use other agents' answers constructively, rather than fully ignoring them.
> > > > > > >
> > > > > > > [ref1]: “Follow the crowd in a new direction: When conformity pressure facilitates group creativity”, Organizational Behavior and Human Decision Processes 2012
> > > > > > >
> > > > > > > [ref2]: “Social influence: Compliance and conformity”, Annual Review of Psychology 2004

---

> > > > > > > ### Author Response · Authors · 2024-11-28
> > > > > > > **Response to Reviewer e696 (2/2)**
> > > > > > >
> > > > > > > Q13: Value of BenchForm.
> > > > > > >
> > > > > > > A13: As outlined in L35-39 of the Introduction, multi-agent systems are increasingly being adopted across various domains, raising concerns about potential unintended consequences, particularly the susceptibility of these agents to cognitive biases akin to those observed in human group dynamics:
> > > > > > >
> > > > > > > > *As these systems evolve, they are increasingly considered for crucial roles in public policy analysis, social platform moderation, and even governance processes. However, the integration of such systems into societal processes raises concerns about potential unintended consequences, particularly the susceptibility of these agents to cognitive biases akin to those observed in human group dynamic.*
> > > > > > > >
> > > > > > >
> > > > > > > From the **experimental design** perspective, BenchForm builds upon and **extends** the design of the Asch conformity experiments to address these concerns. One of the key extensions (also noticed by reviewer ordx) is the study of long-term interactions. From the perspective of **difficulty**, BenchForm proves challenging because: i) mainstream LLMs demonstrate consistent patterns of conformity; and ii) there is currently no solution—neither explored in our paper nor in the rebuttal—that can **perfectly** resolve the issues observed in BenchForm. As a preliminary exploration, we believe BenchForm represents a small yet solid step towards revealing conformity behaviors in multi-agent systems. Moving forward, we will continue to refine the benchmark and explore solutions that may better capture the complexities of real-world multi-agent interactions, as mentioned in Future Work (L517-520).
> > > > > > >
> > > > > > > Q14: Conformity in LLMs *vs* that of humans.
> > > > > > >
> > > > > > > A14: We would like to clarify that our goal is not to impose a direct mapping between human social pressures and LLMs. Rather, we aim to explore the potential for conformity behaviors within multi-agent systems. As mentioned in Limitation, our work did not provide a sufficient condition to prove that LLMs exhibit conformity in the same way humans do. However, our results suggest that, at least to some extent, the answers provided by other agents have a meaningful impact on the subject agent's responses.
> > > > > > >
> > > > > > > We thank the reviewer for your constructive suggestions and look forward to further discussions to refine and expand this work.

---

> > > > > > > > ### Comment · Reviewer_e696 · 2024-12-02
> > > > > > > >
> > > > > > > > Thank you for your hard work and for taking the time to answer all of my questions. However I still stand by my opinion and original score.

---

> > > > > > > > > ### Author Response · Authors · 2024-12-03
> > > > > > > > >
> > > > > > > > > Thank you sincerely for your time and dedication throughout this review process. We truly appreciate your thorough evaluation of our manuscript and rebuttal.

---

### Official Review · Reviewer_kmnc · 2024-10-30

**Soundness:** 3
**Presentation:** 3
**Contribution:** 3
**Rating:** 8
**Confidence:** 3

**Summary:**

This paper addresses the issue of conformity within multi-agent systems based on LLMs (i.e. whether there is a tendency for an LLM-based agent to provide a different response to a question when answering in the context of the answers provided by other agents than when answering the question independently). It starts by presenting a benchmark evaluation environment and empirical results demonstrating both that conformity does arise to a substantial extent in these circumstances, and also that there are large variations in behaviour between different LLMs. It then provides further results highlighting some of the factors which may influence the degree of conformity, as well as results when the agent is asked to reflect on whether/how the answers of others influenced its answer. Finally some preliminary investigations are made into prompt-based methods for encouraging agents to resist pressures to confirm and act as independent thinkers.

**Strengths:**

This is an important topic, and one which I have not personally encountered previously.

The paper provides value in identifying the potential for conformity to arise in multi-agent LLM systems, the problems which might arise from this, and in demonstrating via experimental results that these concerns do actually occur in practice.

The BENCHFORM benchmark and the proposed metrics and testing protocols are a useful starting point for research into the conformity of LLMs. I can see other authors using these tools to build on the results reported in this paper. The 5 interaction protocols were clearly designed, and the purpose of each was clear.

The authors have included multiple LLMs, and multiple variations of each LLM model, within their testing and this has identified that there is a surprising (at least to me) amount of variation in behaviour between these models.

The preliminary results of the mitigation strategies show some promise. Importantly they also demonstrate a variation in outcomes between the different models, highlighting the need for consideration of model-specific issues in future research - it may be that there is no one-size-fits-all solution to the issue of conformity.

The paper was for the most part well structured and written, and the diagrams, tables etc were appropriate and clear.

**Weaknesses:**

At times it is a bit unclear what the desirable behaviour of the agents actually is. This is touched on in Section 7 where the author's comment on the fact that conformity is a double-edged sword - some degree of conformity may assist in consensus-building, but overly conforming agents essentially become useless as they just reinforce the majority answer. I think it would have been useful to include this discussion earlier in the paper, and explain how the proposed metrics relate to this.

The scenarios used in BENCHFORM have some limitations. This is to be expected as this is an initial paper laying the foundations for a new area of research, but I think some of the following issues could have been more clearly addressed in the Limitations and Future Work parts of Section 7.
- The use of multiple choice questions seems like a large simplification of the sort of scenarios where LLMs would be applied.
- The interaction scenarios are effective for the purposes of this initial study, but don't seem to be representative of the way that agents would be allowed to interact in an actual application. If we want the agents to be independent thinkers, why show them the other's answers before they provide their own?
- Following on from that last point, one of the main advantages of building a multi-agent system based on LLMs is that rather than simply using them as an ensemble of models, they could actually be designed to discuss their answers, reason as a group, engage in argumentation etc. I understand why this wasn't done in this study, but this seems like an aspect which needs to be addressed in Future Work.

I was less convinced about the utility of Section 4.3 (the Behavioral Study) compared to the rest of the paper. It's interesting to see the huge divergence in behaviour between the two models reported in Table 4, but does the agent's reporting of its reasoning and the influence of others here actually bear any relationship to the process it followed in providing its original answer, or is it simply a post-hoc rationalization? If its the latter, then using this information as a basis for designing mitigation measures may not be successful.

One minor typo around lines 254/255 - "suppresses" should probably be "surpasses"?

**Questions:**

I'm not familiar with the Asch conformity experiments on which these protocols were based, but the decision to not provide feedback on the correctness of answers (mentioned on page 4) seemed unusual to me. The Trust and Doubt experiments differ only in terms of whether the majority answer provided to the agent is correct or not. In the absence of any feedback on the ground truth, the agent can only learn to trust/doubt its peers based on how frequently it disagrees with their answer (ie working on the assumption that it itself is generally correct). I would have thought a critical factor in learning to trust someone (or vice-versa) is learning through experience that when they disagree with me, they actually tend to be correct. Can you please explain the reasoning behind this aspect of the protocol design?

I found the discussion of Fig 5cd on pages 6 and 7 confusing. It states that Qwen2-72B scores 30.0% for CR^D when the majority is 6, and this increases to 39.9% with a majority of 5. But the graph line showing that behaviour in the figure is the green line, which is labelled as "Trust & CR" (i.e. CR^T). Am I misunderstanding or is there an issue with either the caption or the discussion?

---

> ### Author Response · Authors · 2024-11-22
>
> We thank reviewer kmnc for the valuable time and constructive feedback. We provide point-to-point response below.
>
> Q1: Include the discussion of “conformity is a double-edged sword” earlier.
>
> A1: Good suggestion! We have added this in Sec. 1. (L39~44):
>
> > *… which may manifest in multi-agent systems with both constructive and problematic effects. While conformity can foster consensus and coordination among agents as they interact with humans and each other, …*
> >
>
> In addition, we have added the potential benefit of conformity in Sec. 3.2 (L226~227), as suggested in Reviewer EboB Q1:
>
> > *Note that while $CR^C$ reflects conformity tendencies, this characteristic could be beneficial when LLMs learn from group interactions.*
> >
>
> Q2: Further clarification in the Limitations and Future Work.
>
> A2: Thanks for your suggestion. We have incorporated your insights into Limitations and Future Work:
>
> A2.1: **“The use of multiple choice questions…”:** We agree that the use of multiple-choice questions simplifies the scenarios in which LLMs are typically applied. However, we clarify that our choice to use multiple-choice questions follows the same design of the Asch conformity experiments [ref1-3]. In Asch conformity experiments, participants were also presented with multiple-choice tasks (e.g., choosing the line that matched the length of a reference line). In addition, we have incorporated the following content in Limitations (L506~507):
>
> > *Moreover, the use of multiple-choice questions simplifies the scenarios in which LLMs are typically applied, potentially limiting the generalizability of our findings.*
> >
>
> To help readers better understand our experimental design, we have also included a detailed introduction to the Asch conformity experiments in Appendix §A.
>
> A2.2: **“Interaction in an actual application”:** We recognize that this setup may not fully align with multi-agent interaction scenarios. However, we clarify that the decision to present agents with others' answers before they provide their own also follows the experimental setup of Asch conformity experiments, which are foundational in social psychology for understanding the impact of group pressure on individual behavior. Asch experiments have been widely regarded as a cornerstone in the study of conformity and social influence [ref3-4]. In addition, we have incorporated the following content in Limitations (L504~505):
>
> > *For example, in real-world interactions, agents may not be exposed to others' answers before providing their own.*
> >
>
> A2.3: **“Various collaborative reasoning”:** We have modified the content in Future Work (L517~520):
>
> > *…future work will focus on i) expanding BenchForm to include tasks from the MMLU-Pro dataset and other domains beyond reasoning-intensive problems; and ii) exploring more interaction protocols that better mimic real-world collaborative environments, such as discussing answers, reasoning as a group, engaging in argumentation, etc.*
> >
>
> Q3: Clarify the utility of the behavior study.
>
> A3: Thanks for the insightful comment. Yes, it is hard to determine whether the behavior study reflects genuine reasoning or is merely post-hoc rationalization under the current experimental setup. However, we want to highlight that the empirical results point to two interesting findings: 1) Llama3-70B is more likely to acknowledge conformity and adjust its responses, while Qwen2-72B tends to deny such influence, rarely revising its answers and maintaining confidence. 2) LLMs’ explanations are inconsistent with their decisions. These findings inspire us to develop the “double-checking and reflection” strategy, which is demonstrated to be effective in Sec. 5.2.
>
> Q4: Clarify the decision to not provide feedback on the correctness of answers.
>
> A4: As mentioned in L184~186, the Trust and Doubt protocols draw on the design of the Asch conformity experiments, where participants **are not informed** about the correctness of their answers. Another reason is that providing feedback on the correctness of answers introduces additional variables, potentially confounding the measurement of conformity.
>
> Q5: An issue with the discussion.
>
> A5: Our apologies. You’re right; there is a notation error with the discussion. We have modified the $CR^D$ to $CR^T$ (L337).
>
> Q6: Typos.
>
> A6: Thanks for noticing. It’s fixed.
>
> 1. (L255) “suppresses” → surpasses
>
> We appreciate again your thoughtful review and we hope we addressed your concerns. Please let us know if you'd like any further information.
>
> [ref1] “Effects of group pressure on the modification and distortion of judgments”, Groups, Leadership and Men: Research in Human Relations 1951
>
> [ref2] “Opinions and social pressure”, Scientific American 1955
>
> [ref3] “Studies of independence and conformity. A minority of one against a unanimous majority”, Psychological Monographs 1956
>
> [ref4] “Social influence: Compliance and conformity”, Annual Review of Psychology 2004

---

> > ### Comment · Reviewer_kmnc · 2024-11-22
> >
> > Thank you. I appreciate the changes which you have made.

---

> > > ### Author Response · Authors · 2024-11-22
> > >
> > > Thanks again for your constructive comments! Please let us know if any other concerns.

---

### Official Review · Reviewer_ordx · 2024-11-01

**Soundness:** 4
**Presentation:** 3
**Contribution:** 3
**Rating:** 8
**Confidence:** 3

**Summary:**

In this paper, the authors report on a study aimed at exploring the presence of conformity in Large Language Model (LLM)-driven multi-agent systems. Noting that phenomena like conformity bias and groupthink occur in human groups, the authors hypothesise that something similar might occur in LLMs. To test this, they develop BenchForm. BenchForm consists of reasoning-intensive tasks and five interaction protocols, Raw, Correct Guidance, Wrong Guidance, Trust, and Doubt. BenchForm is used to examine how LLMs perform in groups, specifically whether the LLM adapts its output to better match the group, both in single rounds of discussion and in multiple rounds of discussion. By measuring accuracy, conformity rate and independence rate, the authors identify that, while different LLMs have different characteristics, there is a risk of LLMs conforming their outputs to a majority, even if the output is wrong. Further, they propose that factors influencing conformity include interaction time and majority size and, based on this insight, suggest mitigation strategies such as developing enhanced personas for the LLMs or implementing reflection mechanisms.

**Strengths:**

This is a well-thought out and -presented piece of research.

Originality: It is clearly located within an existing literature where, as the authors note, ‘Research about the phenomenon of conformity within LLM-driven multi-agent systems remains scarce’ (p. 9). One area of originality for the work is that it investigates long-term interactions.

Quality: The way in which the authors draw on sociological research to help develop BenchMark is a good illustration of the importance of bringing insights from different disciplines together, also allowing a novel approach for investigating an underexplored aspect of safety without starting from scratch. This also enhances the soundness of the research. The methodology is clearly described and the various steps are justified.

Clarity: Throughout, the paper is written clearly and follows a logical order that helps to develop the paper’s arguments.

Significance: The authors present findings that suggest that LLMs can be susceptible to forms of conformity bias, with and without recognising the influence on their outputs. In some cases, an LLM may be unwilling to revise a wrong answer even if it has reasoned itself that the answer is wrong (see in particular 4.3 and the tables in the appendix). The paper doesn’t only diagnose a problem, but also offers well-thought through hypotheses for why this might be happening, hypotheses that can potentially explain differences between LLMs, define further avenues of research, and indicate what kind of mitigation strategies could be relevant.

**Weaknesses:**

Overall, I enjoyed the paper. Areas of weakness are areas where I had specific questions arise.

1. Section 4: investigation into factors that influence conformity. Here, the experiments are conducted with Llama3-70B and Qwen2-72B. Why these two? Could the authors explain why these two were chosen as a way of expanding on the methodology used in the set of experiments reported in section 4. Can the authors explain whether there are characteristics of these two LLMs that would make them particularly informative here, or otherwise justify the choice of focusing on these two LLMs.

2. In the original testing, all the additional agents give the same answer. When reading this earlier stage of the paper, I wondered about this particular choice and also whether other scenarios had been explored, e.g. where the majority give one answer but a minority give another answer. This is addressed in section 4.2 on peer pressure where divergent agents are introduced into the studies, but I do think in the description of the original studies some explanation should be made for the initial choice. Could the authors give a brief explanation in the methodology discussion, which I think falls most naturally in section 2.2 on protocols (as the paper is not structured in a standard way with methodology set off in its own section), explaining and justifying why all agents give the same answer in the first iteration of the experiments. To ward off reader questions like mine, it would also be helpful to flag that experiments with divergent agents will be reported on later in the paper, too.

3. Relatedly, does it make a difference if it’s the same agents giving correct/incorrect answers? In other words, is it majority size that matters, or could it be who is in the majority? Perhaps this is an area for further study – the authors note that more work should be done to reflect the varying dynamics of human groups – but I think at the very least some acknowledgement should be made of how these group dynamics could work. The section on peer pressure discussing Asch could be a place to expand on this. I would even welcome some initial experiments being run to see if there is a difference that ought to be explored.

4. In section 5 on empowered personas, I wasn’t sure why adjusting the persona was being introduced as a potential mitigation technique here. Is there anything specific in the findings that suggests that this is a good way to go here, or is it just that this is a route explored already for other issues in the literature? (vs the double-checking and reflection, which seems to be well-motivated by the behavioural study findings.) Some explanation of why this is being presented as a relevant solution, here in the context of the study's findings, would be helpful. For instance, the authors could discuss how the findings on conformity could be addressed through persona adjustments, directly identifying which of the findings make such a mitigation technique have initial plausibility. Alternatively, the authors could clarify that the approach is chosen because of work in other areas.

Typos:
p. 1 ‘that single agent would’ – either ‘a single agent’ or ‘single agents would’
p. 4 ‘multi-gent’ -> ‘multi-agent’
p. 5 ‘suppresses’ – should this rather be ‘surpasses’?
p. 17 A. 1 Data ‘as Table S1 shown’ – should be ‘shows’

**Questions:**

1. Why choose Llama3-70B and Qwen2-72B for the investigation into factors influencing conformity? Are there characteristics of these two LLMs that would make them particularly informative here?

2. Does it make a difference if it’s the same agents giving correct/incorrect answers? Can you run experiments to see if there are differences in scenarios where randomised agents give correct/incorrect answers? (Something like, if there are three agents who give the correct answer 50% or the time and the incorrect answer 50% of the time, but who are part of the majority in the final round.)

3. Is there something specific in the findings that motivate for ‘empowered personas’ as a mitigation strategy?

---

> ### Author Response · Authors · 2024-11-22
>
> We thank reviewer ordx for the valuable time and constructive feedback. We provide point-to-point response below.
>
> Q1: Clarify why Llama3-70B and Qwen2-72B are chosen in Sec. 4 & Sec. 5.
>
> A1: Qwen2-72B is selected because: i) It achieved the highest IR score among the LLMs we tested. ii) An LLM with high IR may demonstrate more varied responses that are not influenced by others' answers, which may facilitate the ablation and behavioral studies. iii) LLMs with higher IR may exhibit unexpected or emergent behaviors, which might spark novel behavior discovery and even lead to discovering new research directions. Llama3-70B is also selected for comparison as it is on the same scale as Qwen2-72B and another reason for selecting Llama3-70B is its widespread use. We have included the reasons for choosing Qwen2-72B and Llama3-70B in the main paper (L293~295):
>
> > *Qwen2-72B is chosen for study due to its achieving the highest IR, which could exhibit diverse or even unexpected responses, potentially aiding in ablation and behavioral studies. Llama3-70B is also selected for comparison, given its comparable scale and widespread usage.*
> >
>
> Q2: Add more contexts about the interaction setting.
>
> A2: Good suggestion! We have added the following content in Sec. 2.2:
>
> > *The additional agents provide the same answer following the experimental setup of Asch conformity experiments. Experiments with divergent opinions are elaborated in §4.2.*
> >
>
> Q3: Majority size *vs* majority composition.
>
> A3: Thanks for sharing such fresh insights! Since the same agents are designed to give correct/incorrect answers in previous discussion rounds, our Trust and Doubt protocols indeed serve as a foundation for examining whether "who is in the majority" matters. The alternative scenario you propose, where agents provide randomized responses in previous rounds, would eliminate the influence of established agent relationships, focusing solely on the majority size. On the other hand, incorporating such randomization in previous rounds introduces additional complexity and hyperparameters, and the metrics designed for the Trust and Doubt protocols can no longer be used. Given these trade-offs, we focus on varying the majority size within our Wrong Guidance and Correct Guidance protocols. This provides a proper approximation and yields credible results.
>
> We have added the results in Appendix §D. The results show that the impact of majority size is minimal on Llama3-70B but significant on Qwen2-72B. We conclude that both majority size and majority composition influence conformity, but the extent of their impact varies depending on the LLM.
>
> Q4: Clarify the motivation for “empowered personas”.
>
> A4: Throughout the community, assigning LLMs distinct "personas" has been increasingly recognized as an effective means to enhance system performance. In multi-agent systems, giving each agent a distinct role is known to improve coordination and increase efficiency. [ref1] demonstrated that commercial AI systems often define the role of LLMs through system prompts, and [ref2] highlighted the growing trend of embedding personas in LLMs to simulate human-like behavior. In our initial setup, we constrained the LLM to the persona of “a helpful assistant,” which limits its potential and does not fully explore the broader capabilities of persona-driven interactions.
>
> Q5: Typos.
>
> A5: Our apologies. They are now corrected.
>
> 1. (L50) “that single agent would” → that a single agent would
> 2. (L192) “multi-gent” → multi-agent
> 3. (L255) “suppresses” → surpasses
> 4. (L884) “Data as Table S1 shown” → Data as Table S1 shows
>
> We appreciate again your thoughtful review and we hope we addressed your concerns. Please let us know if you'd like any further information.
>
> [ref1] When "A Helpful Assistant" Is Not Really Helpful: Personas in System Prompts Do Not Improve Performances of Large Language Models, EMNLP 2024.
>
> [ref2] Bias Runs Deep: Implicit reasoning biases in persona-assigned LLM, ICLR 2024.

---

> ### Comment · Reviewer_ordx · 2024-11-22
>
> Thanks for these explanations and changes. I'm happy with them. However, I would still encourage including something in the text to explain the motivation for using 'empowered personas'. Just because an approach has wider backing does not yet show that it is appropriate for addressing a particular problem, and not making one's motivations clear risks obscuring how general trends could nevertheless hide a mismatch. Put another way, in the paper you've got: 'Previous works [51–53] on LLMs’ roles suggest that adjusting LLMs’ persona via system prompts can enhance *specific capabilities* and lead to better performance'. But why suppose that adjusting personas is relevant for the *specific capabilities* involved in mitigating conformity?
>
> If your motivation is simply that the wider community recognizes that personas are 'an effective means to enhance system performance', then that should be stated. However, it seems relevant for the problem at hand that personas are used to simulate human-like behaviour, and so the motivation (and justification) is stronger than a simple appeal to current practice. Even just including the response you've got here in the paper, I think, would be sufficient to address this comment.

---

> > ### Author Response · Authors · 2024-11-22
> >
> > Thank you so much for your response.
> >
> > We want to clarify that the motivation for using empowered personas stems originally from Finding III from Sec. 3.3, which reveals that different LLMs exhibit distinct behavioral characteristics. For example, Qwen2-7B demonstrates high credulity (L273-275), and the Llama3.1 series exhibit significant resistance to external guidance (L283-284). These contrasting behaviors inspire us to design personas that balance these tendencies, enabling LLMs to make nuanced decisions without being overly trusting or excessively resistant. We agree that clear motivations for using 'empowered personas' is of great importance, and have added the relevant content in Sec. 5.1:
> >
> > > *The motivation for using empowered personas stems from two core observations: i) Finding III in §3.3 shows that LLMs struggle to make independent decisions. ii) Previous works [51–53] suggest that adjusting LLMs’ persona can enhance specific capabilities.*
> > >
> >
> > We would like to thank you again for your valuable comments. Please let us know if any other concerns.

---

> > > ### Comment · Reviewer_ordx · 2024-11-25
> > >
> > > Thanks, I do think this is an important addition.

---

> > > > ### Author Response · Authors · 2024-11-25
> > > >
> > > > Thanks again for your thoughtful review! Please let us know if any other concerns.

---

### Official Review · Reviewer_EboB · 2024-11-06

**Soundness:** 4
**Presentation:** 4
**Contribution:** 3
**Rating:** 8
**Confidence:** 4

**Summary:**

Authors investigated conformity in LLMs, examining how they may change their responses due to group influence, similar to conformity bias and groupthink in human interactions. Through a new introduced benchmark, BENCHFORM, authors evaluate conformity across four protocols and 11 LLMs. The study explored three main aspects: the existence of conformity and independence rates, factors that influence conformity (such as interaction steps, and majority size), and potential strategies to mitigate conformity effects.  They proposed two mitigation strategies: 1) enhanced personas and 2)reflection mechanism to mitigate the influence. The results provide insights that could support the development of more robust and ethically aligned AI collaborations.

**Strengths:**

* Presentation: The paper is well-written and is easy to follow.
* Benchmark: Introduction of BENCHFORM offers a unique framework for studying conformity in LLM-driven agents.
* Experimentation: The empirical studies are through providing great findings about possible ways that LLMs can be influenced, bringing attention to the ethical and policy challenges ahead.

**Weaknesses:**

* While I agree ${CR}^C$ also shows conformity, but if it helps LLMs to learn from the forum it can be beneficial. I recommend including that aspect in the paper.
* 272: Flipping between Qwen2-72B and Qwen2-7B in the result section was confusing. I recommend either including both in all results are stick to one.

Minor:
* 223: To note that -> Note that,
* 389: Can you add statistical significance on the chart?

**Questions:**

336: Why do you think increasing population from 5->6 increased doubt conformity but not trust conformity?

**Details Of Ethics Concerns:**

The findings of this paper can be misused by practitioners to conform LLMs to provide incorrect information. While authors did a great job surfacing such challenges, it might be worthwhile for extra pair of eyes to review it.

---

> ### Author Response · Authors · 2024-11-22
>
> We thank reviewer EboB for the valuable time and constructive feedback. We provide point-to-point response below.
>
> Q1: Include the potential benefit of $CR^C$.
>
> A1: Agree! We have added the following content in Sec. 3.2 (L226~227):
>
> > *Note that while $CR^C$ reflects conformity tendencies, this characteristic could be beneficial when LLMs learn from group interactions.*
> >
>
> Q2: Include both Qwen2-72B and Qwen2-7B in all results or stick to one.
>
> A2: Good suggestion! We add the result of Qwen2-72B and the discussion about it in Finding III (L280~281):
>
> > *However, Qwen2-72B exhibits characteristics of a more independent thinker. Its CRs under the three protocols are only about 30%, with only $CR^T$ reaching 56%.*
> >
>
> Q3: Add statistical significance on Fig. 6 & 7.
>
> A3: Thanks for your reminder. We have incorporated statistical significance on Fig. 6 & 7.
>
> Q4: Why did increasing majority size (5→6) raise doubt but not trust conformity?
>
> A4: For the Doubt protocol, increasing the majority size raises conformity, which is intuitive. In terms of the Trust protocol, increasing the majority size from 5 to 6 lowers conformity. We hypothesize that introducing one single dissenting voice may reinforce, rather than weaken, the subject agent's tendency to side with the majority (as mentioned in L336~338).
>
> We appreciate again your thoughtful review and we hope we addressed your concerns. Please let us know if you'd like any further information.

---

> > ### Comment · Reviewer_EboB · 2024-11-24
> >
> > Appreciate the detailed responses.

---

> > > ### Author Response · Authors · 2024-11-24
> > >
> > > Thanks again for your constructive comments! Please let us know if any other concerns.

---

### Author Response · Authors · 2024-11-22
**Summary of Revisions**

To all reviewers:

Thank you so much for your careful review and suggestive comments. We have revised our paper according to your comments and uploaded the **updated paper** to OpenReview. The major changes are as follows:

1. We add experiments for comparing majority size *vs* majority composition according to Reviewer ordx’s comments in Appendix §D.
2. We add experiments for LLMs with the best CR according to Reviewer e696’s comments in Appendix §G.
3. We add experiments in finetuning Llama3-8B according to Reviewer ordx’s comments in Appendix §H.
4. We discuss the potential benefit of $CR^C$ according to the comments of Reviewer EboB and kmnc in L226~227.
5. We modify Finding III to include both Qwen2-72B and Qwen2-7B according to Reviewer Ebob’s comments in L280~281.
6. We add statistical significance on Fig. 6 & 7 according to Reviewer Ebob’s comments.
7. We clarify why Llama3-70B and Qwen2-72B are chosen in Sec. 4 & Sec. 5 according to the comments of Reviewer ordx and Reviewer e696.
8. We add more contexts about the interaction setting in L186~188 according to Reviewer ordx’s comments.
9. We add more details about the Asch conformity experiments in Appendix §A according to Reviewer kmnc’s comments.
10. We add further clarification in the Limitations and Future Work according to Reviewer kmnc’s comments.
11. We correct all typos according to the comments of Reviewer ordx and kmnc.

Sincerely yours,

Authors.

---

### Meta-Review · Area_Chair_wUWB · 2024-12-20

**Metareview:**

All authors and myself were all in agreement that this paper is an important and timely project to study conformity in large language models. The paper takes inspiration from the famous Asch conformity experiments and shows that a similar phenomenon occurs with LLMs, something I myself had recently been independently wondering about too. I'm glad to see that these effects have now been demonstrated with LLMs.

**Additional Comments On Reviewer Discussion:**

This paper was not terribly controversial. All the reviewers basically liked it, and the conversation did not uncover any serious issues.

---

### Decision · Program_Chairs · 2025-01-22

Accept (Oral)